# Adaptive DDoS detection mode in software-defined SIP-VoIP using transfer learning with boosted meta-learner

Rume Elizabeth Yoro[1], Margaret Dumebi Okpor [2], Maureen Ifeanyi Akazue[3], Ejaita Abugor Okpako[4], Andrew Okonji Eboka[5], Patrick Ogholuwarami Ejeh[6], Arnold Adimabua Ojugo[6], Chris Chukwufunaya Odiakaose[7], Amaka Patience Binitie[5], Rita Erhovwo Ako[6], Victor Ochuko Geteloma[6], Paul Avwerosuo Onoma[6], Asuobite ThankGod Max-Egba[6], Ayei Egu Ibor[8], Sunny Innocent Onyemenem[5], Elochukwu Ukwandu[9]*

**1** Department of Cybersecurity, Dennis Osadebey University, Asaba, Delta State, Nigeria, **2** Department of Cybersecurity, Delta State University of Science and Technology Ozoro, Ozoro, Delta State, Nigeria, **3** Department of Computer Science, Delta State University, Abraka, Delta State, Nigeria, **4** Department of Computer Science, University of Delta, Agbor, Delta State, Nigeria, **5** Department of Computer Education, Federal College of Education (Technical), Asaba, Nigeria, **6** Department of Computer Science, Federal University of Petroleum Resources Effurun, Effurun, Nigeria, **7** Department of Computer Science, Dennis Osadebey University, Asaba, Delta State, Nigeria, **8** Department of Computer Science, University of Calabar, Calabar, Cross Rivers State, Nigeria, **9** Department of Applied Computing, Cardiff School of Technologies, Cardiff Metropolitan University, Wales, United Kingdom

* eaukwandu@cardiffmet.ac.uk

## Abstract

The Internet has continued to provision its infrastructure as a platform for competitive marketing, enhanced productivity, and monetization efficacy. However, it has become a means for adversaries to exploit unsuspecting users and, in turn, compromise network resources. The utilization of filters, gateways, firewalls, and intrusion detection systems has only minimized the effects of adversaries. Thus, with the constant evolution of exploitation and penetrative techniques in network security, security experts are required to also evolve their mitigation and defensive measures by using advanced tools such as machine learning approach(es) poised to help detect and stop as close to its source, any attack or threat. This will help to quickly identify malicious packets and prevent resource exploits and service disruption. To curb these, studies have sought to minimize the effects of these attacks via advanced machine learning (ML) inspired tools. Traditional ML performance is often degraded due to: (a) its simplistic design that is unsuitable to handle categorical datasets effectively, and (b) its adoption of hill-climbing mode that yields solution(s) that are stuck at local maxima. To avoid such pitfalls, we use deep learning (DL) schemes based on recurrent networks. They present the demerits of the vanishing gradient problem and require longer training time. To curb the challenges of ML and DL, we propose a transfer learning scheme with 3-base (BiGRU, BiLSTM, and Random Forest) classifiers and XGBoost meta-learner to aid effective identification of DDoS. The ensemble

**Data availability statement:** Dataset was retrieved from https://data.mendeley.com/datasets/b7vw628825/1 – array of DDoS traffic attack in a software-defined network.

**Funding:** The author(s) received no specific funding for this work.

**Competing interests:** The authors have declared that no competing interests exist.

yields Accuracy and F1 of 1.000 to effectively classify 314,102-DDoS-cases during its evaluation. The proposed ensemble demonstrates that it can efficiently identify malicious packets for DDoS attacks in network transactions.

---

## 1. Introduction

Businesses and corporate organizations have become more vigilant in their commitment to mitigating threats associated with cybersecurity as global losses run into Billions of Dollars [1]. Despite these, attackers have continued to evolve techniques poised at circumventing secure protocols [2,3]. With the efforts of using secure gateways and firewalls to enhance user trust and access to network resources yielding less than expected results [4], stakeholders need to reposition using approaches that will accurately identify adversarial exploits in this multi-billion dollar global battle [5]. All the same, as businesses can initiate-establish-terminate an online call via Session Initiation Protocol (SIP) on a Voice Over Internet Protocol (VoIP) telephony [6] to facilitate efficient Peer-to-Peer (P2P) communication, IP spoofing has become commonplace. IP spoofing has become a common security flaw in SIP-based VoIP networks [7], as an adversary can assume the identity of a legitimate user to create access points targeted at exploiting unsuspecting users. For instance, using a man-in-the-middle attack, false messages can be triggered, which impacts the integrity of a SIP/VoIP call. By also listening to a legitimate interaction, an adversary can pose as a legitimate user and attack the interaction via a subterfuge attack, either via flooding or probing [8,9].

The proliferation of social media platforms has spurred new frontiers for attacks [10], with users' suspicion levels now elevated with more online presence. Sentiments have proven to be an integral facet of a user's personality traits that drives desires [11]. Attack designs can utilize: (a) believability feat that increases possibility of a device to accept malicious data as originating from a genuine user [12], and (b) insidiousness measures the rate at which a malicious content remains potent and undetectable to the user device [13]. Threats by design are poised to weaken networks by obscuring data privacy and evading security by presenting themselves as legitimate users [14]. This is achieved via intrusive acts, service outage, and denial to a user [15]. The exponential rise of these attacks has been attributed to the broad range of availability in the constructive technology itself and calls for effort by security experts and stakeholders in mitigating these threats by exploring various protocols [16,17]. Socially engineered threats against network resources continue to witness a high success rate as users are repeatedly compromised due to personality traits [18,19]. The ease with which these attacks are propagated, such as Distributed Denial of Service (DDoS), has become of great concern to businesses [20], even with the rise in the number of tools and techniques to mitigate such attacks [21,22].

### 1.1 DDoS attacks on SIP VoIP-based infrastructure

These type of attacks target user devices to derail networked resources [23] from their original purposes and can sometimes utilize social-engineered tools in

harvesting users' credentials. These credentials are then used to compromise a network infrastructure [24] and resources such as memory, Central Processsing Unit (CPU) time, and network bandwidth [25,26]. In many cases, an adversary can achieve this feat via carefully coordinated and crafted exploits that insert obfuscated (malware) as requests that are poised to overwhelm a network. The magnitude of the exploits depends on the size of the explored botnet, which unveils the severity of the threat [27]. DDoS are carefully crafted threats that flood a network server with user requests and exploit the targeted network resources by denying legitimate users access to services [28,29]. As a first aid measure, manually disconnecting a (detected) compromised device is a common approach to fix the challenge; and once compromised, the device becomes an adversary's entry point to proceed on other targets within the network infrastructure for further exploits [30]. Detection schemes for DDoS attacks can be grouped into: (a) victim-end detection [31,32] (b) core-end detection [33–35], and (c) source-end detection [36–38].

DDoS attacks are grouped into: (1) **spoofing**: adversary sends large volume of malicious packets to a server by spoofing/masking its source-IP address – making it tedious to differentiate between genuine and malicious packets [39], and (2) **flooding**: adversary floods a network with user requests that exhausts network resources – making it difficult for legitimate users to access the network resources [40]. Flood-based attacks flood available (network) resources with massive amounts of user requests that eventually create a spoofed packet traffic [41]. This inadvertently blocks legitimate user devices from being serviced with the available infrastructure resources. An exploitative DoS variant is the distributed DoS that originates from multiple sources, making the flood-based DoS a menace to VoIP infrastructure with loss of monetization [42]. The last network architecture layer aids data transfer with programs (i.e., browsers, email, etc) that provide services such as FTP, IMAP, Telnet, SMPT [43,44], and so on. Susceptible targets of DDoS attacks, especially on SIP-based networks, aim to disrupt SIP proxy services or their network users [45,46].

## 1.2. Learning schemes for DDoS detection

Identification tasks are grouped into: machine learning (ML), deep learning (DL), and ensemble learning (EL) [47]. MLs as used in high-dimension tasks are trained to identify hidden relations of interest in (un)structured datasets to support decisions in the quest for truth [48]. Their robustness, reusability, and flexibility help them learn such relations quickly as changes occur via feature engineering to ease outlier identification in the functioning of a system [49]. Thus, it determines crucial predictors selected for model construction as input and, in turn, recognizes those to aggregate as output. For classification cum regression tasks, the use of ML scheme are poised to help identify cum recognize hidden relations between the underlying predictor features of interest – as the model/heuristic seeks to learn changes within the dataset as means to support decisions in its quests for ground-truth [50]. For these, researchers often explore common traditional MLs such as Random Forest [51], SVM [52], Naïve Bayes [53,54], etc. However, most traditional MLs: (a) explore/utilize hill-climbing techniques that often traps such solutions at local minima – making them not optimally fit, (b) traditional ML simplistic design and nature often yields degraded performance as they may not be able to effectively handle categorical dataset, and (c) they are not robust enough to handle large dataset [55]. For these and other reasons, researchers then explore deep learning (DL) schemes. DL are networks tailored to capture underlying relations of interest in a dataset [56]. Its vanishing gradient challenge impedes performance and hinders the widespread use of Recurrent Neural Network (RNN) [57]). Its variant, the Long-Short-Term Memory (LSTM), resolves this challenge by exploring input-gates that effectively manage how quickly and easily the model adapts to changes observed in the dataset [58,59]. A major degrade to LSTM is its need for longer training time and inability to handle large datasets [60,61].

To combat the challenges in both ML and DL, EL fuses both ML and DL using ML to overcome the issues in DL and vice versa. Thus, the EL yields a single and stronger optimal fit classifier. This feat is achieved via: (a) vote, (b) bagging, (c) boost, and (d) stacked schemes. In vote mode, classifier(s) are independently aggregated to yield a final output with enhanced performance. While it does rely on their fused predictive relations. This unexplored fusion degrades performance if more diversity and outliers exist in the dataset [62]. Bagging trains similar decision trees with equal vote weights

(s). It minimizes the variance and bias in a dataset by randomly training its tree with k-fold train-data so that the model aggregates all tree predictions to yield greater accuracy with reduced errors [63,64]. With boost, it sequentially trains independent decision trees so that each iteration yields a classifier that corrects the mistakes of its base (previous) learners in the output [65]. Thus, with each iteration, the ensemble learns and amends its predecessors' errors to yield enhanced performance with ADAboost as an example. Lastly, the stacked mode explores transfer-learning mode, which trains its (meta) learner to efficiently fuse the predictive outcome of its base-classifiers to improve the generalization performance of its (meta)classifier. This flexibility yields enhanced outcomes with less convergence time and fewer iterations.

This fusion will benefit us as thus: (a) the stacked learning will exploit the benefits of the various base schemes explored to devoid the model of overfit while resolving gaps in the categorical/complex dataset [66], and (b) the XGB-regressor will leverage and boost the predictive capabilities inherent the stacked learning to enhance itself, more profitably. Boosting effectively improves the performance of its learners by improving the difficulty witnessed with its previous iterations, which in turn, improves its outcome as well as reduces both variance and bias in the dataset. Thus, the ensemble will lean on the comprehensive knowledge of the various approaches to exploit the learning depths of its base models with a boosted (meta-learner) mode that yields improved error reduction. But, performance is often degraded due to the imbalanced nature of the dataset [67]. To yield a balanced distribution, studies have explored oversampling schemes (as opposed to undersampling that makes increasingly meaningless, the chosen dataset) [68]. A common oversample mode used is SMOTE (synthetic minority oversample technique) [69], and its variant SMOTE-Tomek [70] fuses a SMOTE (over-sampler) with Tomeks (undersampler) [71] as a means to reduce its data distribution class-overlap, improve the dataset's quality, and yield fastened learning.

### 1.3. Study motivation

The study is motivated as thus [72–74]: (a) limited availability/access to right-quality datasets to aid model construction, training and evaluation [75], (b) imbalanced nature of datasets where attacks (minor-class) transactions trails behind genuine (major-class) transactions [76], (c) rise in multi-cross-channel transactions, newer schemes must account for it as data acquisition mode to enhance model's performance and keep up with emergent tactics [77,78], and (d) quest for adaptive detection scheme against VoIP-based DDoS flood-attacks in software-defined network must be poised to identify low data traffic rate(s) [79–81], stealthy traffics, abrupt, high surges and flash-crowd flood-attacks [82]. To minimize this, a model is embedded with targeted delivery on a gateway to ensure the network application layer is protected from all incoming and outgoing packets [83].

Thus, we capture dynamic predictor feats via the trial-and-error mode in pursuance of a model that yields an optimal solution and satisfies the target class with improved generalization and devoid of model overfit. This study proposes to fuse ML and DL with SMOTE-Tomek sampling via transfer (boosted) learning ensemble for robust DDoS attack identification and classification. The study contributes as thus: (a) utilizes SMOTE-Tomek to improve data quality and distribution, (b) develops three (i.e., BiLSTM, BiGRU, and Random Forest) base learners, (c) fuses the base-learners via transfer learning with XGBoost as a meta-learner, and (d) evaluates the proposed method with popular SDN-DDoS test datasets to prove the method's robustness.

## 2. Materials and methods

Our proposed method as shown in Fig 1 adopts the stacked learning mode with the following steps:

1. **Step 1 – Data Collection**: Dataset was retrieved from https://data.mendeley.com/datasets/b7vw628825/1 – an array of DDoS traffic attacks in a software-defined network. With 26 features and 1,048,575 records as in Table 1, it yields minor-class 328,765 (attacks) and major-class 719,810 (genuine cases). Fig 2 is a plot of the original data distribution. The input data are read and stored in the data frame.

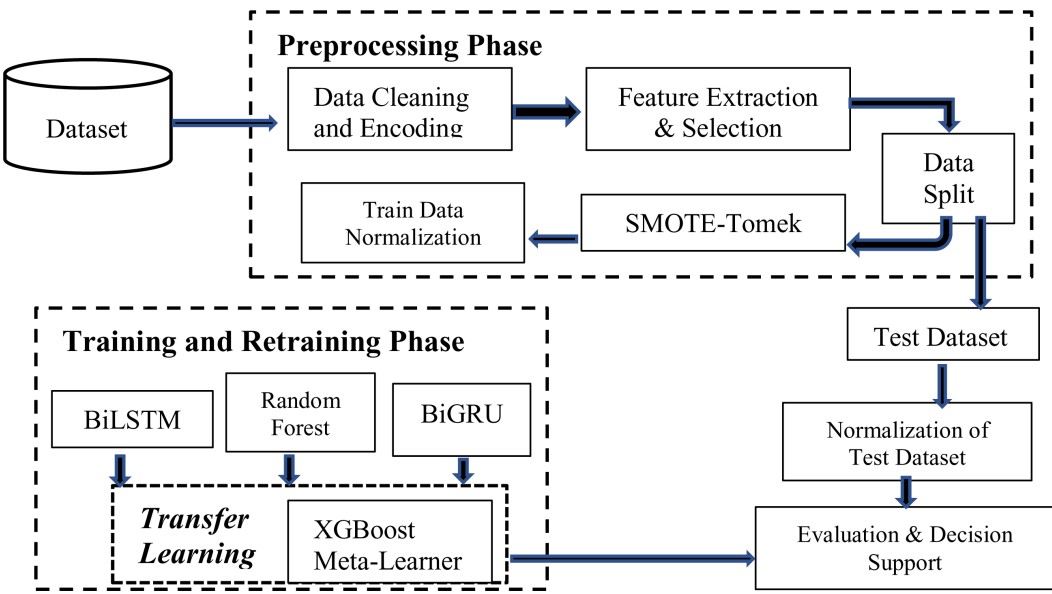

**Fig 1. Proposed Stacking Ensemble Approach with XGBoost as Meta-heuristics.**

2. **Step 2 – Preprocessing:** Here, we perform the actions of cleaning as follows: (a) remove duplicate records to ensure the dataset is devoid of record redundancies, (b) remove missing values to ensure data quality, and (c) to yield an optimized, restructured dataset [84] distributed into a variety of labeled-classes. We then encode the records using the one-hot encoding technique, which helps the heuristic to transform all categorical data into their binary equivalent for effective utilization by the ML heuristic. A detailed code listing is as in algorithm listing 2.1.

```
Algorithm Listing 2.1

Import pandas as pd; import matplotlib,pyplot as plt; import seaborn as sns #import Python libraries
1. df = pd.read_csv('files/data/sdn_csv') #loads the software-defined network dataset
2. df.head(10) #shows the maximum of 10 samples from the chosen dataset
   #data cleaning
3. plt.figure(figsize = (10,6)) sns.heatmap(df.isna().transpose(), cmap='YlGnBu", cbar_kws={'label':
   'Missing Data'})
   #handling the missing values
4. def data_imputation(data, column_group, column_selected):
5. group = data[column_group].unique()
6. for value in group:
7.   median = data.loc[(data[column_group]==value) & ~(data[column_select].isna()), column_select].
     median() #get median value
8.   median=data.loc[(data[column_group]==value) & ~(data[column_select].isna()), column_select].
     median #change miss value
9. return data #endfor and return the dataframe
10.sns.boxplot(df['source_IP'])
   #change the missing values
11.condition_de = (df['source_ip'].notnull())
12.df['source_IP'] = df['source_IP'].mask(condition.de, df['source_IP']: sns.boxplot[df['source_IP']
   #remove duplicate data values
13.df = df.drop_duplicates().reset_index(drop=True)
```

This yields an optimized dataset and allows the proposed system to initiate the feature selection process.

**Table 1. Ranking of Attributes score using the Chi-Square.**

| Features | Format | Data Types | $R^2$-Value | Selected (Yes/No) |
|---|---|---|---|---|
| Source IP | a.b.c.d | Object | 10.041 | Yes |
| Source Port | a.b.c.d | Object | 9.956 | Yes |
| Destination IP | Numeric | Integer | 10.001 | Yes |
| Destination Port | Numeric | Integer | 9.248 | Yes |
| Port_number | Numeric | Float | 2.470 | No |
| Protocol | String | Object | 8.492 | Yes |
| tx_bytes | Numeric | Integer | 5.372 | No |
| rx_bytes | Numeric | Integer | 4.222 | No |
| Duration | H:M:S | Float | 9.258 | Yes |
| Total_duration | H:M:S | Float | 3.029 | No |
| Flows | Numeric | Integer | 1.891 | No |
| Pair_flow | Binary | Integer | 3.092 | No |
| Packet_per_message | Numeric | Integer | 6.929 | No |
| Packet_count | Numeric | Integer | 5.759 | No |
| Packets_per_flow | Numeric | Float | 3.561 | No |
| Packet_rate | Numeric | Float | 3.364 | No |
| Packet_loss | Numeric | Float | 0.419 | No |
| byte_count | Numeric | Integer | 1.956 | No |
| Bytes_per_flow | Numeric | Float | 3.012 | No |
| Total_kilo_bytes | Numeric | Integer | 0.248 | No |
| Delay | Numeric | Float | 2.471 | No |
| Jitters | Numeric | Float | 8.492 | No |
| label_attack | Binary | Integer | 9.372 | Yes |

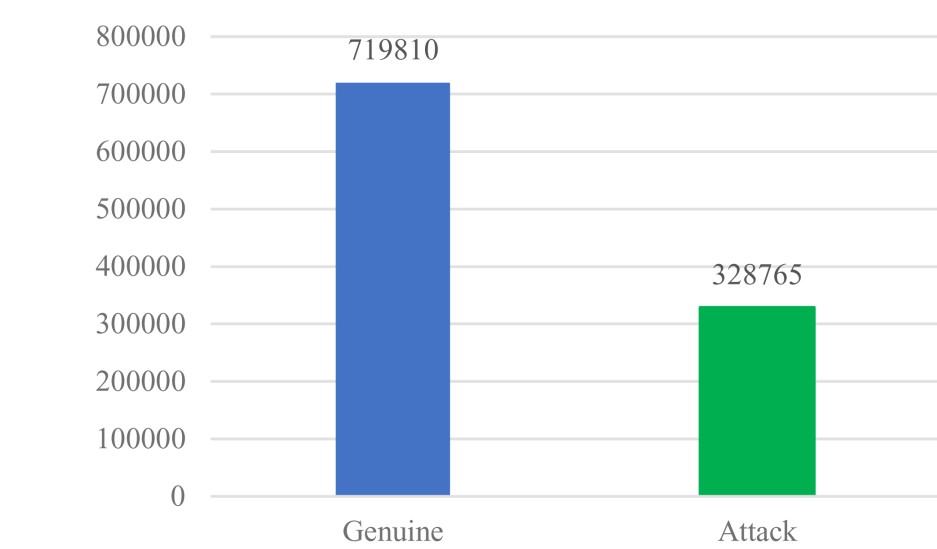

**Fig 2. Original Dataset plot.**

3. **Step 3 – Feature Selection** extracts data label(s) as predictor variables, assessing which labels are pinned down as input (x) vis-à-vis choosing label that the proposed model will forecast as its output (y). Feature selection evaluates and assesses which predictors yield important relations in the quest for ground-truth (target class). It discards predictor(s) that yield no significance (irrelevant or docile) in the quest. Thus, it yields reduced dimensionality in the chosen predictors, both quickens the construction of the heuristics as well, and improves training to yield enhanced generalization. Studies have continued to posit that this is especially useful for model construction, where cost is a crucial characteristic in the quest for our target class [85,86]. Lastly, the model is evaluated on how fit and/or close to optimal the selected features correlate to a target class. Thus, we utilize the relief ranking function (for the feature selection approach) as in Equation 1 and detailed in Algorithm Listing 2.2. This was used to compute the resulting threshold alongside the feature ranking values (by importance) for each predictor about the target class, as seen in Table 1. With the threshold set at 8.321, a total of 7 predictors (i.e., features) were selected with requisite in the quest for anomaly detection (i.e., target-class 1).

$$Y = 100 * \sum \left| \left(x_1^2 - x_2^2\right)^2 + \left(1 - x_1^2\right)^2 \right| \tag{1}$$

Algorithm listing 2.2 is a step-by-step, relief ranking feature selection mode [87].

---
**Algorithm Listing 2.2**

---
```
1. With dataset: n ← number of train samples), a ← number of features), m ← random train samples
   used to update W
2. initialize all feature weights W[A]=0.0
3. for i = 1 to m do:
4.   randomly select a target instance R
5.   find nearest hit 'H' and nearest miss 'M' (instances)
6.   for A = 1 to m do:
7.   W[A] = W[A] – diff(A,R,H)/m + diff(A,R,M)/m
8.   end for: end for
9. return vector W of feature scores that estimate the quality of features
```
---

The 4th and 5th columns of Table 1 yield the ranking of all data labels (and attributes) as contained therein the dataset, which were scored using the Chi-Square feature ranking for the various predictor features.

4. **Step 3 – Dataset Split and Balancing:** First, the dataset is split into train (70%) and test (30%) subsets, to allow for balancing to be actioned only on the train dataset. With the dataset chosen, data points are grouped into minor and major class distributions. Thus, balancing causes a resampling of these data labels via the actions of nearest-neighbour interpolation to either remove some labels or create synthetic instances that repopulate the pool to yield a redistributed, and more balanced distribution within the major-and-minor classes. Here, we adapt **SMOTE-Tomek** links mode via SMOTE (oversample) and Tomek-links (under-sampler) scheme achieved as thus: (a) it samples the original dataset pool, identifying both the major-and-minor classes, (b) it then creates synthetic labels for the minor-class, while removing (under-samples) labels from majority-class closest to the minority-class [70], and (c) these newly created synthetic labels are added to the original pool to yield a more balanced class(es) distribution as in Figs 2 and 3 respectively. While algorithm 2.3 details the step-by-step approach, the SMOTE-Tomek links approach to data balancing for the task.

---
**Algorithm Listing 2.3**

---
```
Input: M(minor_class_sample); N(synthetic_sample); number_k_nearest_neighbor for i in range(N);
1. from minor_class, choose random data-point//start SMOTE_mode
2. compute: relative_distance from randomly_selected_data and k_nearest_neighbor
3. choose rnd_val = random_value(0,1): rnd_val * relative_distance;
```
---

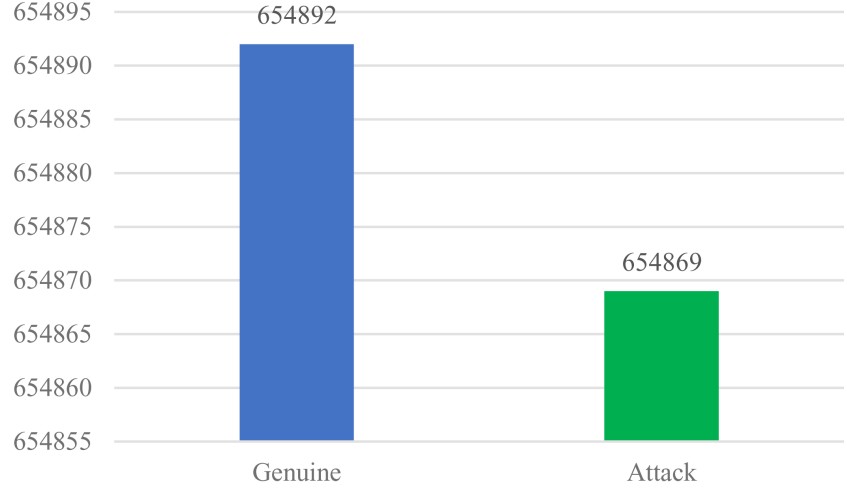

**Fig 3. Dataset with SMOTE applied.**

```
4. if simulated_samples = obtained then minor_class_new = minor_class + simulated_samples
5. repeat steps 2-to-4 until threshold_minor_class_new = reached;
6. select rnd_minor_class(data)//start Tomek_Links (under-sampler) approach
7. find k_nearest_neighbor(randomized_data)
8. if k_nearest_neighbor.selected = minor_class_new then TomekLink created
9. stop TomekLink procedure: end
```

5. **Step-5 – Normalization** via feature transformation, reengineers the imbalanced dataset to resample the class-distribution(s). We use the standard-normalizer function as in Equation 2, which selects data-labels from the resampled dataset to yield a distribution mean of 0, and deviation of 1, with $\mu$ as mean, z as our normalizer, x as the original data, and $\sigma$ as standard deviation. Fig 4 shows the normalized plot. Afterwards, for the study, the dataset is split into a 70% training dataset and a 30% test dataset.

$$z = \frac{(x - \mu)}{\sigma}$$
(2)

6. **Step-6 – Transfer (Stacked) Learning Ensemble Design** – leans on 3-base models with XGBoost meta-classifier as in the proposed methodology, as thus:

✓ The Bidirectional Long Short-Term Memory (BiLSTM) is based on the RNN model, useful for handling time-series datasets. The RNN yields a gradient vanishing problem, such that its gradient for the learning process becomes quite small. This slows down or eventually stops all learning within the model. LSTM overcomes this challenge via utilization of gates (i.e., input, forget, and output), which effectively allows the network to learn when to 'recall' and when to 'forget' irrelevant knowledge. In addition, its cell state update function ($C_t$) maintains all important knowledge over the period and is not impaired or degraded by the vanishing gradient problem. The gates are constructed using the Equation (3)-(5), respectively:

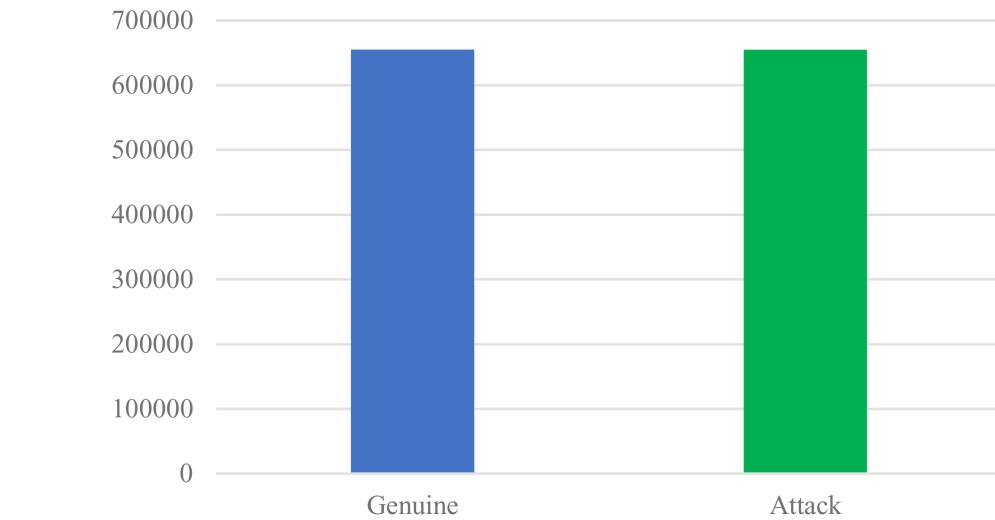

**Fig 4. Normalized Dataset after prior SMOTE-Tomek Links.**

$$i_t = \sigma\left(W_i\left[h_{t-1}, x_t\right] + b_i\right) \tag{3a}$$

$$f_t = \sigma\left(W_f\left[h_{t-1}, x_t\right] + b_f\right) \tag{3b}$$

$$o_t = \sigma\left(W_f\left[h_{t-1}, x_t\right] + b_o\right) \tag{3c}$$

$$\overline{C}_t = tanh\left(W_c\left[h_{t-1}, x_t\right] + b_c\right) \tag{4a}$$

$$C_t = f_t * C_{t-1} + i_t * C_t \tag{4b}$$

$$h_t = o_t * \tanh(C_t) \tag{5}$$

With $i_t$ as activation of the input gate, $o_t$ is an activation of output gate, $\sigma$ is sigmoid function, $W_f$ is the weight of the forget gate, $h_{t-1}$ is the hidden state from previous timestamp, $x_t$ is input at time t, $b_f$ is the bias for forget gate, $\overline{C}_t$ is the candidate value for the memory cell, and $h_{t-1}$ is the hidden state at time t. Additionally, BiLSTM, as a variant of the LSTM, can process data using forward or backward formations. Its first layer allows the flow of data in one direction (i.e., source to destination), while the second layer reverses the flow of data (from destination to source) so that the network possesses past and future context of data-series [70]. This paradigm is useful and expressive in natural language processing. The BiLSTM proffers greater flexibility via fusion of knowledge from both directions (s). It carefully utilizes hyperparameters that tune the model to avoid slow convergence, model overfit, memory efficiency, and task distribution. This is seen in Table 2.

**Table 2. BiLSTM Design and Model configuration with Hyper-predictor tuning.**

| Predictor Settings | Value(s) | Description |
|---|---|---|
| RNN_layer | Bidirectional (LSTM(64)) | Bidirectional RNN: 64 LSTM (first layer) and 32 LSTM (second layer) |
| retun_sequence | True (for first layer) | Returns the entire output sequence for the first layer |
| input_shape | x_train_scaled.shape [1], 1 | Same length as number of predictors in x_train_scaled with one feat per timestep |
| dense_layer | y_train_resampled_max() + 1 | Layer has the same units as classes in y_train_resampled/ output_layer |
| activation_dense_layer | Softmax | Activation function used in the output for multi-class classification |
| optimizer | Adam | learning_rate=0.001, beta_1=0.9, beta_2=0.999, epsilon-1e-07 |
| loss_function | categorical_crossentry | Loss function for multi-class classification |
| metrics | accuracy | Metrics upon which the model is evaluated during training and retraining |

✓The Bidirectional Gated Recurrent Units (BiGRU) – The LSTM without proper setting is caught up with the challenge of the vanishing problem. BiGRU yields a simpler structure (as a variant RNN) [88] and also overcomes the vanishing gradient problem in LSTM as it fuses both the input and forget gates into a single update gate; And in turn, reduces the number of predictors to be trained. This speeds up the construction of the model and its training without trading off much of its memory capability [89]. Similar to BiLSTM, the BiGRU yields a 2-way data processing capability to capture the before/after context in each data sequence. It achieves this via the Update and Reset gates as in Equations (6)-(7), respectively.

$$u_t = \sigma\left(W_u\left[h_{t-1}, x_t\right]\right) \tag{6a}$$

$$r_t = \sigma\left(W_r\left[h_{t-1}, x_t\right]\right) \tag{6b}$$

$$\overline{h}_t = tanh\left(W\left[r_t * h_{t-1}, x_t\right]\right) \tag{7a}$$

$$h_t = \left(u_t * h_{t-1}\left(1 + u_t\right) * \overline{\overline{h}}_t\right) \tag{7b}$$

Where $u_t$ as update gate, $\sigma$ is sigmoid function, W is weight matrix, $W_u$ is weight of update gate, $h_{t-1}$ as hidden state in previous time, $x_t$ is input at time t, $r_t$ is reset gate, $\overline{\overline{h}}_t$ is new hidden state candidate value for the memory cell, and $h_t$ is the updated hidden state at time t. Thus, the model captures bidirectional data context to yield an improved understanding of all intricate data dependencies with a carefully tuned hyper-predictor to help achieve greater balance for train speed, result convergence, memory requirements, enhanced accuracy, and task distribution. Model design and configuration are seen in Table 3.

✓The Random Forest (RFE) as a supervised learning, tree-based model, utilizes the bagging approach to grow its decision trees independently. Each tree is constructed using the bootstrap aggregation, which explores the majority vote

**Table 3. BiLSTM Design and Model configuration with Hyper-predictor tuning.**

| Predictor Settings | Value(s) | Description |
|---|---|---|
| RNN_layer | Bidirectional (GRU(64)) | Bidirectional RNN: 64 GRU units (first layer) and 32 GRU units (second layer) |
| retun_sequence | True (for first layer) | Returns entire output sequence for the first layer |
| input_shape | x_train_scaled.shape [1], 1 | Same length as number of predictors in x_train_scaled with one feat per timestep |
| dense_layer | y_train_resampled_max() + 1 | Layer has the same units as classes in y_train_resampled/ output_layer |
| activation_dense_layer | softmax | Activation function used in the output for multi-class classification |
| optimizer | adam | learning_rate=0.001, beta_1=0.9, beta_2=0.999, epsilon-1e-07 |
| loss_function | categorical_crossentry | Loss function for multi-class classification |
| metrics | accuracy | Metrics upon which the model is evaluated during training and retraining |

that samples the training data during prediction [90]. In addition, it provides an extra layer that extends/changes how the decision trees are constructed as a means to reintroduce randomness with a binary-tree split on each node. Thus, its best predictor nodes are randomly selected via its recursive structure to capture intrinsic interactions between parameters (data-labels). Its major demerit, however, is its flexibility and robustness with complexity and mutation as contained therein the dataset [91,92], causing degraded performance [74,93]. To curb this, we tune the RF hyperparameters to help address dataset imbalance and diversity, reduce overfitting of the ensemble, and also yield improved performance accuracy. Table 4 shows the RF ensemble design and configuration.

✓ The XGBoost meta-regressor, like the Random Forest, is a tree-based ensemble that exploits the gradient boost approach to identify labels in a dataset. It aggregates the sum of its weaker, base learners via a series of iterations to yield a fit solution [94]. Thus, for each new iteration and corresponding outcome, the XGBoost corrects the weaknesses of its base learners to yield an improved ensemble. This is achieved via its goal function, which minimizes its loss

**Table 4. Random Forest Design and Configuration with Hyperparameter Tuning.**

| Predictor Settings | Value(s) | Description |
|---|---|---|
| n_estimators | 150 | Number of trees constructed |
| min_weight_fraction_leaf | 0.1 | The tree's structure is based on the weight assigned to each sample |
| learning_rate | 0.25 | Step size learning for update |
| eval_metric | error, logloss | Performance evaluation metrics |
| random_state | 25 | The seeds for reproduction |
| max_features | 5 | Maximum number of features to construct the RF tree ensemble |
| min_sample_split | 10 | Minimal samples needed |
| eval_set | x,val, y_val | Train data for evaluation |
| max_depth | 5 | Max depth of each tree |
| min_sample_leaf | auto | Number of predictors to be used and considered |
| bootstrap | True | Ensures bootstrap aggregation use |

function as in [95,96]. Ensemble can also be tuned to address dataset imbalance and diversity, reduce overfitting of the ensemble, and also yield improved performance accuracy [97,98]. Table 5 shows the XGBoost design and configuration.

7. **(Re)Training** aims to fit the proposed ensemble with a SMOTE-Tomek, normalized training dataset with a 10% training dataset applied for retraining (or cross-validation). With the stratified k-fold-dataset poised to yield (with each fold) a good representation of the dataset, the stacked learning ensemble ensures the heuristic proffers enhanced generalization that is devoid of overfit. During (re)training – the sample rule(s) as generated are explained as thus, with Table 6 showing the top-18 rules:

*if* (duration="-1:0:23", protocol="telnet" and src-port= -1, dest-port=23, srcIP="192.168.1.30", dest-IP ="192.168.0.20) *then* {log network connection as an **Intrusion**}.

**Table 5. XGBoost Design and Configuration with Hyperparameter Tuning.**

| Predictor Settings | Value(s) | Description |
|---|---|---|
| n_estimators | 250 | Number of trees constructed |
| learning_rate | 0.25 | Step size learning for update |
| eval_metric | error, logloss | Performance evaluation metrics |
| random_state | 25 | The seeds for reproduction |
| eval_set | x,val, y_val | Train data for evaluation |
| max_depth | 5 | Max depth of each tree |

**Table 6. Top 18-Generated Rules ranked with attributes.**

| Time | Protocol | Source Port | Destination Port | Source IP | Destination IP | Attack | Fitness |
|---|---|---|---|---|---|---|---|
| −1,0,23 | telnet | −1 | 23 | 192.168.1.30 | 192.168.0.20 | PG | 0.8063 |
| 0,0,5 | −1 | −1 | −1 | 192.168.1.30 | 192.168.0.20 | PS | 0.8063 |
| −1,0,23 | telnet | −1 | 23 | 192.-1.1.30 | 192.168.0.20 | PC | 0.8063 |
| 0,0,5 | −1 | −1 | −1 | 192.168.1.30 | 192.168.0.20 | ARS | 0.8063 |
| −1,0,23 | telnet | −1 | 23 | 192.168.1.30 | 192.168.0.20 | ICMP | 0.8063 |
| 0,0,5 | −1 | −1 | −1 | 192.168.1.30 | 192.168.0.20 | NP | 0.8063 |
| 0,0,23 | telnet | −1 | −1 | 192.168.1.30 | 192.168.0.20 | PA | 0.8063 |
| −1,0,23 | telnet | −1 | 23 | 192.168.1.30 | 192.168.0.20 | FA | 0.8063 |
| −1,0,23 | telnet | −1 | 23 | 192.168.1.30 | 192.168.0.20 | ARS | 0.8063 |
| 0,0,-1 | −1 | 1023 | 1021 | 192.-1.1.30 | −1.168.0.20 | PODA | 0.8031 |
| **−1,0, 1** | **−1** | **1023** | **−1** | **192.168.1.30** | **192.168.0.-1** | **PODA** | **0.8031** |
| 0,0,14 | −1 | −1 | 513 | 192.168.1.30 | 192.168.0.20 | SR | 0.8031 |
| 0,0,14 | −1 | −1 | 513 | −1.168.1.30 | 192.168.0.20 | SH | 0.8031 |
| 0,0,14 | −1 | −1 | 513 | 192.168.1.30 | 192.168.0.-1 | RA | 0.8031 |
| −1,0,-1 | −1 | 1023 | −1 | 192.168.1.30 | 192.168.0.-1 | DN | 0.8031 |
| 0,0,5 | −1 | −1 | 23 | 192.168.1.30 | 192.168.0.20 | IPS | 0.8031 |
| −1,0, 1 | −1 | 1023 | −1 | 192.168.1.30 | 192.168.-1.20 | PODA | 0.8031 |
| 0,0,14 | −1 | −1 | 513 | 192.168.1.30 | 192.168.0.-1 | ICMP | 0.8031 |

# 3. Results and analysis

## 3.1. Results and findings

Table 7 shows performance evaluation metrics for all base-learners (Random Forest, BiGRU, and BiLSTM) with the XGB-regressor (meta-learner). The XGB meta-learner is used to readily resolve the conflict generated by the diversity of the fused heuristics and inherent encoding complexities in the dataset. Thus, the model is devoid of overfit as the transfer learning approach combines the predictive capability of all 3-base classifiers.

Both BiLSTM and BiGRU outperformed the Random Forest with accuracy of (RF, BiLSTM, and BiGRU) as 0.9815, 0.9968, 0.9981; Recall of 0.9745, 0.9848, and 0.9881; Precision of 0.9805, 0.9318, and 0.9541, and F1 of 0.9805, 0.9881, and 0.9925, respectively. Our stacked meta-learner yielded 1.000 for Accuracy, Recall, F1, and Precision, respectively. Our simplistic stacked (transfer) learning design yields reduced computational complexities, reduced overhead, and enhanced performance. Our near-perfect F1 and Accuracy allow for more integration from its base classifiers, as XGB's regularization term efficiently moderates ensemble overfit for more accurate, generalizable predictions in practical implementation. Proposed ensemble efficiently reduced skewed variance and bias inherent in the dataset, to yield a more robust and stable ensemble for new data and hidden variables inherent in the training dataset.

Fig 5 yields the training loss for the proposed scheme. It yields a consistent, significant decrease from 0.69 in the first epoch to 0.31 by the third epoch. This trend signifies that the proposed scheme successfully minimizes errors on the training dataset. A smooth, monotonic decrease in loss without sudden fluctuations or plateaus indicates that the model's learning rate is well-tuned and that it is not encountering instability or vanishing gradients. Conversely, Fig 6 shows validation accuracy per epoch with a continuous rise from 0.71 to over 0.90 across the three epochs. It implies how well

**Table 7. Stacking Learner Evaluation with base/met learner performances.**

| Features | Accuracy | Precision | Recall | F1 |
|---|---|---|---|---|
| Random Forest | 0.9815 | 0.9805 | 0.9745 | 0.9805 |
| BiLSTM | 0,9968 | 0.9318 | 0.9848 | 0.9881 |
| BiGRU | 0.9981 | 0.9541 | 0.9881 | 0.9925 |
| **Meta-Learner** | | | | |
| XGBoost | 1.0000 | 1.0000 | 0.9999 | 1.0000 |

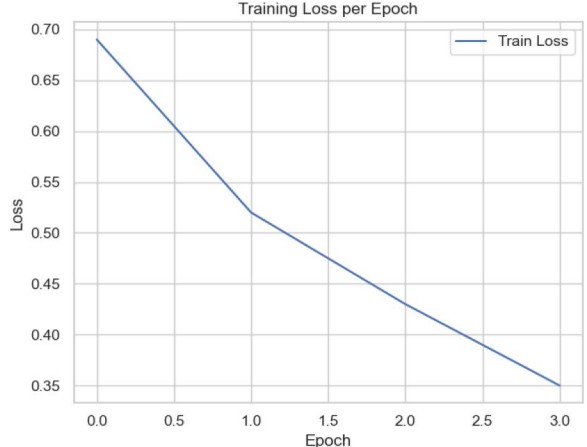

**Fig 5. Training Loss for the proposed model.**

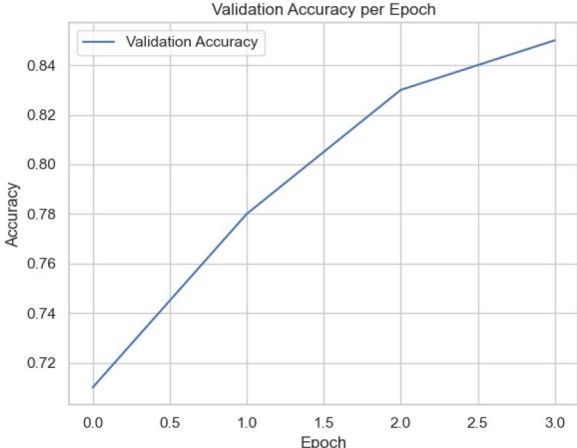

**Fig 6. Validation Accuracy for the proposed model.**

the scheme performs with the unseen data not used during its training. This improvement suggests that the model is not merely memorizing the training data (overfit); Rather, it learns the intricate features as captured in the test dataset that generalize effectively to new SIP traffic samples. The upward trajectory implies a healthy learning process and demonstrates that the model can reliably detect SIP-VoIP DDoS flood-based attacks with increasing accuracy, harmonic mean, precision, and recall [99].

Our proposed ensemble effectively identifies DDoS attack data accurately and has proven to efficiently reduce bias and variance indicative of the confusion matrix as in Fig 7, yielding a more stable, robust model for new data and/or hidden underlying parameters of interest within a domain's training dataset being considered. Our study supports that SMOTE-Tomek proffered greater influence in the quest for ground-truth and impacted the overall performance by identifying features of importance that influence prediction. Model effectively classified genuine from malicious packets with perfect accuracy and with perfect F1-score to accurately classify all 314,573 instances, which agrees with [100,101].

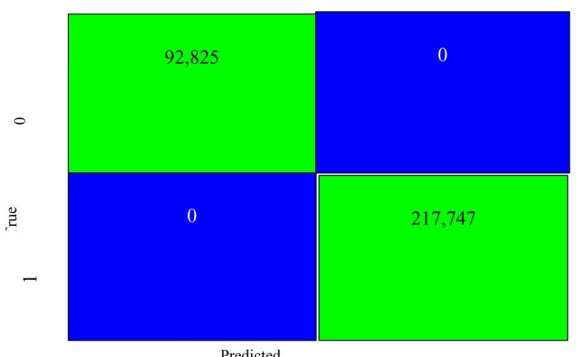

**Fig 7. Confusion matrix for the stacking ensemble.**

**Table 8. Benchmarking and Comparative Testing of Proposed Stacking Ensemble.**

| Authors with Date | Approach | F1 | Accuracy | Precision | Recall |
|---|---|---|---|---|---|
| Ref [15] | BiGRU deep learning | 1.0000 | 1.0000 | 0.9999 | 1.0000 |
| Ref [104] | LR, AdaBoost, SVM, KNN | 0.7902 | 0.7815 | 0.7372 | 0.7025 |
| Ref [105] | Multi-dimensional deep convolution classifier (MDDCC) | 0.9881 | 0,9968 | 0.9848 | 0.9318 |
| Ref [106] | CNN-LSTM | 0.9925 | 0.9981 | 0.9881 | 0.9541 |
| Ref [107] | CNN-BiGRU-AM (attention mechanism) | 0.9890 | 0.9840 | 0.9715 | 0.9620 |
| Our Method | BiLSTM-RF-BiGRU with XGB-regressor | 1.0000 | 1.0000 | 1.0000 | 0.9999 |

### 3.2. Comparative analysis

As we explored the high performance of our proposed ensemble across the dataset to demonstrate its flexibility, adaptability, robustness, and prediction ability, we also benchmarked it against previous methods that have utilized the same dataset as seen in Table 8 [102,103].

Whilst some task datasets have proven much easier to recognize/classified, Others have also conversely proven to be more painstaking. Some domain task(s), such as medical and image records, require that their chosen ensemble design metric is strongly impacted by the consequence of diagnostic errors within the captured dataset. Thus, the measure of both specificity and sensitivity becomes 2 critical feats to be evaluated since they are directly related to their inherent outcomes [94].

### 3.3. Practical Implementation and Implications

For our target system delivery, we tested the SMOTE-Tomek fused stacked-based learning ensemble as an embedded application program interface on a web standalone program via Flask. The Flask is a lightweight Python framework that easily integrates as an embedded app, as incorporated with Streamlit to provide the requisite platform to transform this spam detection ensemble into an accessible API. This, in turn – yielded a Fast-API deployed with 3-components: (a) *initialize* function specifies opened communication ports, (b) *integrate* function connects the API framework to the server system – and thus, allow the processing as filter of all incoming packets, and (c) *interoperability* function processes all messaging data from/to all the interconnected devices.

## 4. Conclusion

Our proposed model yields a total of 60 rules, with the top 18 rules found to have a classification accuracy range [0.89, 0.98]. This stresses and advances the evidence that over 89% of the generated rules can adequately identify cum classify the DDoS dataset. This ideology and paradigm of many good-fit rules is far-reaching and better than achieving a single elitist rule. This, in turn, increases the chances of recognizing malicious packets. Also, the nature of attacks in network transactions requires a constant concerted effort of all to detect intrusion. ML-based schemes simply sniff across network requests, analyze their traffic pattern (anomaly detection), and decide which transactions are compromised.

## Author contributions

**Conceptualization:** Rume Elizabeth Yoro, Margaret Dumebi Okpor, Maureen Ifeanyi Akazue, Andrew Okonji Eboka, Patrick Ogholuwarami Ejeh, Arnold Adimabua Ojugo, Chris Chukwufunaya Odiakaose, Amaka Patience Binitie, Rita Erhovwo Ako, Victor Ochuko Geteloma, Ayei Egu Ibor.

**Data curation:** Rume Elizabeth Yoro, Maureen Ifeanyi Akazue, Ejaita Abugor Okpako, Patrick Ogholuwarami Ejeh, Sunny Innocent Onyemenem.

**Formal analysis:** Margaret Dumebi Okpor, Ejaita Abugor Okpako, Andrew Okonji Eboka, Chris Chukwufunaya Odiakaose, Rita Erhovwo Ako, Victor Ochuko Geteloma, Paul Avwerosuo Onoma, Asuobite ThankGod Max-Egba.

**Funding acquisition:** Rume Elizabeth Yoro, Margaret Dumebi Okpor, Andrew Okonji Eboka, Patrick Ogholuwarami Ejeh, Arnold Adimabua Ojugo, Amaka Patience Binitie, Rita Erhovwo Ako, Victor Ochuko Geteloma, Ayei Egu Ibor, Sunny Innocent Onyemenem.

**Investigation:** Rume Elizabeth Yoro, Arnold Adimabua Ojugo, Amaka Patience Binitie, Ayei Egu Ibor.

**Methodology:** Maureen Ifeanyi Akazue, Ejaita Abugor Okpako, Sunny Innocent Onyemenem.

**Project administration:** Andrew Okonji Eboka, Arnold Adimabua Ojugo, Elochukwu Ukwandu.

**Resources:** Arnold Adimabua Ojugo, Victor Ochuko Geteloma, Paul Avwerosuo Onoma, Asuobite ThankGod Max-Egba, Ayei Egu Ibor.

**Software:** Maureen Ifeanyi Akazue, Patrick Ogholuwarami Ejeh, Chris Chukwufunaya Odiakaose.

**Validation:** Rume Elizabeth Yoro, Margaret Dumebi Okpor, Ejaita Abugor Okpako, Sunny Innocent Onyemenem, Elochukwu Ukwandu.

**Writing – original draft:** Rume Elizabeth Yoro, Margaret Dumebi Okpor, Maureen Ifeanyi Akazue, Ejaita Abugor Okpako, Andrew Okonji Eboka, Patrick Ogholuwarami Ejeh, Chris Chukwufunaya Odiakaose, Amaka Patience Binitie, Rita Erhovwo Ako, Victor Ochuko Geteloma, Ayei Egu Ibor, Sunny Innocent Onyemenem.

**Writing – review & editing:** Rume Elizabeth Yoro, Arnold Adimabua Ojugo, Rita Erhovwo Ako, Paul Avwerosuo Onoma, Asuobite ThankGod Max-Egba, Elochukwu Ukwandu.

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
