## [Decision Letter · Decision Letter 0]

Dear Dr. Ukwandu,

Thank you for submitting your manuscript to PLOS ONE. After careful consideration, we feel that it has merit but does not fully meet PLOS ONE’s publication criteria as it currently stands. Therefore, we invite you to submit a revised version of the manuscript that addresses the points raised during the review process.

We look forward to receiving your revised manuscript.

Kind regards,

Taimur Bakhshi

Academic Editor

PLOS ONE

Journal Requirements:

Journal of Fuzzy Systems and Control, Vol. 2, No 3, 2024 ISSN: 2986-6537, DOI: 10.59247/jfsc.v2i3.267

https://unidel.edu.ng/cms/uploads/publications/unidel_pub_1733046009.pdf?

In your revision ensure you cite all your sources (including your own works), and quote or rephrase any duplicated text outside the methods section. Further consideration is dependent on these concerns being addressed.

Additional Editor Comments:

Please revise and resubmit the manuscript according to reviewers' comments. Thank you.

Reviewers' comments:

Reviewer's Responses to Questions

**Comments to the Author**

1. Is the manuscript technically sound, and do the data support the conclusions?

Reviewer #1: Partly

Reviewer #2: Yes

Reviewer #3: Partly

2. Has the statistical analysis been performed appropriately and rigorously?

Reviewer #1: Yes

Reviewer #2: Yes

Reviewer #3: Yes

3. Have the authors made all data underlying the findings in their manuscript fully available?

Reviewer #1: Yes

Reviewer #2: Yes

Reviewer #3: Yes

4. Is the manuscript presented in an intelligible fashion and written in standard English?

Reviewer #1: Yes

Reviewer #2: Yes

Reviewer #3: No

Reviewer #1: - Abstract: The abstract should succinctly summarize the objectives, methods, key results, and conclusions of the study. Ensure it clearly outlines the significance of the research.

- Introduction: Strengthen the framing of the research question. Provide a more comprehensive literature review to justify the study's necessity, outlining gaps that the research addresses.

- Study Design: Clearly define the study design. When applicable, include controls and randomization methods to enhance the reliability of the findings.

- Sample Size: Justification of the sample size through power analysis should be included to demonstrate the adequacy of the study to detect statistically significant differences.

- Data Collection: Describe in detail how data were collected, including any instruments or questionnaires used, and their validation processes. This allows for replication and verification of the results.

- Statistical Methods: Clearly outline the statistical methods used for data analysis, including assumptions, and provide a rationale for choosing those methods. If applicable, ensure that results are presented with appropriate confidence intervals and p-values.

- Results Presentation: The results section should be structured clearly, often using tables and figures for better readability. Each visual should include a descriptive caption and must be referenced in the text.

- Interpretation of Findings: Contextualize findings with respect to existing literature. Address any contradictions with previous studies and provide explanations.

- Limitations: Explicitly acknowledge potential limitations of the study. Discuss how these limitations might affect the interpretation of results.

- Ensure the conclusion succinctly returns to the significance of the findings and offers clear recommendations for practitioners and suggestions for future research.

- All cited works should be current and relevant. Ensure that the referencing format is consistent with the journal’s guidelines, which can be crucial for publication.

- Ensure that all figures and tables are of high quality and have appropriate legends. This visual data must complement the textual descriptions and provide clarity.

While the manuscript outlines a technically sound approach utilizing transfer learning for DDoS detection and presents data supporting its findings, a full assessment would require more detailed information about the experimental controls, replication, and sample sizes used. The mention of using well-established classifiers and the dataset of 314,102 instances indicates a rigorous framework, but specifics on those experimental aspects would clarify the robustness of the conclusions drawn.

The manuscript mentions utilizing methods such as SMOTE-Tomek for data balancing and highlights performance metrics like accuracy and F1-score for evaluation. However, without more detailed information regarding the specific statistical methods used, their assumptions, controls, and thoroughness of the analysis, it is difficult to fully ascertain the rigor of the statistical analysis. Additional clarity on these points would provide a better evaluation of the appropriateness of the statistical analysis performed.

Reviewer #2: This manuscript falls within the scope of the journal but several comments and suggestions that the manuscript should address:

1. Clearly articulate the novel contributions of the proposed ensemble compared to existing DDoS detection approaches, especially the role of transfer learning with a boosted meta-learner.

2. Expand the related work section to include a deeper comparison with alternative methods for SIP-VoIP DDoS detection and highlight how this approach advances the state‐of‐the‐art.

3. Provide more details about the dataset, including its limitations, distribution of classes, and any potential biases introduced during data collection or preprocessing.

4. Clarify the rationale behind choosing the specific base classifiers (BiLSTM, BiGRU, and Random Forest) and explain how their complementary strengths improve overall detection performance.

5. Describe in detail the hyperparameter tuning process (e.g., grid search, cross-validation) for all components, including the ensemble’s meta-learner.

6. Address concerns of potential overfitting given the near-perfect performance metrics; consider incorporating additional validation methods or independent test datasets.

7. Discuss the impact of the SMOTE-Tomek balancing technique compared with other data augmentation methods and justify its selection..

Reviewer #3: Review of the Paper: "Adaptive DDoS Detection Mode in Software-Defined SIP-VoIP Using Transfer Learning with Boosted Meta-Learner"

The paper proposes a transfer learning-based DDoS detection model for SIP-VoIP using BiGRU, BiLSTM, Random Forest, and XGBoost. While the topic is relevant and the approach promising, several areas require major revision to enhance clarity, justification, and practical applicability.

Key Revisions Needed:

• Classifier Justification: Explain why BiGRU, BiLSTM, and Random Forest were chosen over other models. Compare with alternative deep learning classifiers.

• Dataset & Preprocessing: Provide details on feature selection, normalization, and handling of class imbalance. Clarify preprocessing steps.

• Related Work: Expand comparison with existing SIP-VoIP DDoS detection models and highlight novelty.

• Results & Explainability: Discuss false positives, false negatives, and model interpretability. Include confusion matrices, ROC curves, or SHAP analysis to strengthen insights.

• Abstract & Conclusion: Add specific performance metrics and emphasize practical implications. Ensure clear takeaways.

• Practical Deployment: Address real-time implementation, IDS integration, and robustness against adversarial attacks.

• Grammar & Readability: Improve structure, clarify research gaps, and refine writing for better comprehension.

Recommendation: Major Revision

The paper presents a strong concept but requires significant improvements in methodology, dataset details, result interpretation, and writing clarity before it can be considered for publication.

**Do you want your identity to be public for this peer review?** For information about this choice, including consent withdrawal, please see our Privacy Policy

Reviewer #1: No

Reviewer #2: No

Reviewer #3: No

---

## [Author Response · Author response to Decision Letter 1]

31 Mar 2025

Review of the Paper: "Adaptive DDoS Detection Mode in Software-Defined SIP-VoIP Using Transfer Learning with Boosted Meta-Learner"

The paper proposes a transfer learning-based DDoS detection model for SIP-VoIP using BiGRU, BiLSTM, Random Forest, and XGBoost. While the topic is relevant and the approach promising, several areas require major revision to enhance clarity, justification, and practical applicability.

Key Revisions Needed:

• Classifier Justification: Explain why BiGRU, BiLSTM, and Random Forest were chosen over other models. Compare with alternative deep learning classifiers. Comments: As address in Section 1.2

• Dataset & Preprocessing: Provide details on feature selection, normalization, and handling of class imbalance. Clarify preprocessing steps. Comments: Pre-processing steps are detailed with the formulae for computation as in step-3 of Section 2 ‘Materials and Methods’. Data Balancing scheme as executed in Step-3 is detailed as suggested. Normalization is achieved in Step-5 using the standard normalizer of Equation 1.

• Related Work: Expand comparison with existing SIP-VoIP DDoS detection models and highlight novelty. Comments: As address in Section 1.2.

• Results & Explainability: Discuss false positives, false negatives, and model interpretability. Include confusion matrices, ROC curves, or SHAP analysis to strengthen insights. Comments: Model interpretability addressed in Section 3.1.

• Abstract & Conclusion: Add specific performance metrics and emphasize practical implications. Ensure clear takeaways. Comments: Revised Abstract

• Practical Deployment: Address real-time implementation, IDS integration, and robustness against adversarial attacks. Comments: See the Conclusion section for the implementation of the ensemble method as an embedded system application programming interface (API)

• Grammar & Readability: Improve structure, clarify research gaps, and refine writing for better comprehension.

---

## [Decision Letter · Decision Letter 1]

PLOS ONE

Dear Dr. Ukwandu,

Thank you for submitting your manuscript to PLOS ONE. After careful consideration, we feel that it has merit but does not fully meet PLOS ONE’s publication criteria as it currently stands. Therefore, we invite you to submit a revised version of the manuscript that addresses the points raised during the review process.

**Please update the manuscript according to reviewer's comments and have this re-submitted. Thank you.**

We look forward to receiving your revised manuscript.

Kind regards,

Dr. Taimur Bakhshi

Academic Editor

PLOS ONE

Journal Requirements:

Additional Editor Comments:

You are requested to resubmit an updated manuscript, considering the reviewers comments on priority. Thank you.

Reviewers' comments:

Reviewer's Responses to Questions

**Comments to the Author**

Reviewer #1: All comments have been addressed

Reviewer #2: All comments have been addressed, some additional items have been highlighted.

Reviewer #3: All comments have been addressed, some minor items have been highlighted.

2. Is the manuscript technically sound, and do the data support the conclusions?

Reviewer #1: Yes

Reviewer #2: (No Response)

Reviewer #3: Yes

3. Has the statistical analysis been performed appropriately and rigorously?

Reviewer #1: Yes

Reviewer #2: (No Response)

Reviewer #3: Yes

4. Have the authors made all data underlying the findings in their manuscript fully available?

Reviewer #1: Yes

Reviewer #2: (No Response)

Reviewer #3: Yes

5. Is the manuscript presented in an intelligible fashion and written in standard English?

Reviewer #1: Yes

Reviewer #2: Yes

Reviewer #3: Yes

Reviewer #1: - You need to provide a more thorough justification for choosing BiGRU, BiLSTM, and Random Forest over other potential models. Did you compare their performance with alternative deep learning classifiers? Including comparative analysis or reference results can strengthen your argument for your chosen classifiers.

- While you reference where preprocessing details can be found, you should explicitly state your feature selection criteria, normalization techniques, and methods for handling class imbalance directly in the main text. Make it easy for readers to understand each step without having to refer to other sections extensively.

- Expand upon the existing literature in the field of SIP-VoIP DDoS detection. Current citations seem limited. To clarify your contribution and novelty, you should discuss how your approach differs from or improves upon previous work.

- A deeper analysis of your results is required. Discuss the implications of any false positives and false negatives you encountered and how they affect the model's reliability. Including visual aids such as confusion matrices and ROC curves is essential here, along with a discussion on model interpretability and potential avenues for improvement.

- In the Abstract, be sure to include specific performance metrics such as accuracy, precision, recall, and F1-score of your models. In your conclusion, highlight the practical implications of your work and how it can be applied in real-world scenarios, ensuring clear takeaways for the reader.

- Address how your DDoS detection model can be implemented in real-time and integrated into existing IDS frameworks. Discuss its robustness against adversarial attacks more comprehensively. This section should convince readers of the feasibility of your approach in a practical setting.

- Focus on improving the overall structure and flow of the writing. Ensure technical jargon and terminology are clearly defined upon first use and maintain consistency throughout. This will aid comprehension for readers who may not be familiar with the subject matter.

- If you have visual data representation (e.g., graphs or tables), ensure they are properly labeled and referenced in the text. Each figure/table should have a clear legend or caption explaining what it represents.

Reviewer #2: The manuscript is under journal scope but need to some modifications should be addressed as follows:

1. Provide a detailed justification for selecting these classifiers, including a comparison with alternative deep learning or machine learning models. Highlight their strengths and relevance to the problem being addressed.

2. Describe the dataset in detail, including its size, source, and characteristics.

3. Include confusion matrices, ROC curves, precision-recall curves, or SHAP (SHapley Additive exPlanations) analysis to provide deeper insights into model performance.

4. Update the abstract to include key performance metrics (e.g., accuracy, F1-score, AUC-ROC).

5. Discuss the robustness of the system against adversarial attacks and propose strategies to enhance security.

6. Refine the language to ensure clarity and readability.

Reviewer #3: (No Response)

**Do you want your identity to be public for this peer review?** For information about this choice, including consent withdrawal, please see our Privacy Policy

Reviewer #1: No

Reviewer #2: No

Reviewer #3: No

---

## [Author Response · Author response to Decision Letter 2]

19 May 2025

Reviewer #1: - You need to provide a more thorough justification for choosing BiGRU, BiLSTM, and Random Forest over other potential models. Did you compare their performance with alternative deep learning classifiers? Including comparative analysis or reference results can strengthen your argument for your chosen classifiers. Comments: The utilization of Machine Learning (ML) schemes as data-driven model aids researchers to glean insightful knowledge for a domain task. However, a major issue with such models includes: (a) the more complex the model – the better its improved generalization and performance especially in handling large complex dataset, and (b) many of these traditional ML schemes such as Random Forest, SVM, AdaBoost, LR, GA, etc – have been known not to effectively handle categorical data. With these reasons as in Section 1.2 – researchers exploit deep learning (DL) scheme such as RNN, LSTM, BiLSTM, BiGRU, etc, as methods to address these inherent challenges with traditional ML.

- While you reference where preprocessing details can be found, you should explicitly state your feature selection criteria, normalization techniques, and methods for handling class imbalance directly in the main text. Make it easy for readers to understand each step without having to refer to other sections extensively. Comments: Addressed in Section 2 – Materials and Methods (with step-2) as marked red.

- Expand upon the existing literature in the field of SIP-VoIP DDoS detection. Current citations seem limited. To clarify your contribution and novelty, you should discuss how your approach differs from or improves upon previous work. Comments: Literature Review addressed in Section 1.2 – while, contributions and novelty addressed in Section 1.3. It details how this study improves the existing works via its contributions as marked red.

- A deeper analysis of your results is required. Discuss the implications of any false positives and false negatives you encountered and how they affect the model's reliability. Including visual aids such as confusion matrices and ROC curves is essential here, along with a discussion on model interpretability and potential avenues for improvement. Comments: Addressed in Section 3.1.

- In the Abstract, be sure to include specific performance metrics such as accuracy, precision, recall, and F1-score of your models. In your conclusion, highlight the practical implications of your work and how it can be applied in real-world scenarios, ensuring clear takeaways for the reader. Comments: Addressed in Abstract.

- Address how your DDoS detection model can be implemented in real-time and integrated into existing IDS frameworks. Discuss its robustness against adversarial attacks more comprehensively. This section should convince readers of the feasibility of your approach in a practical setting. Comments: Addressed in Section 3.1 Result and Findings.

- Focus on improving the overall structure and flow of the writing. Ensure technical jargon and terminology are clearly defined upon first use and maintain consistency throughout. This will aid comprehension for readers who may not be familiar with subject matter. Comments: Addressed.

- If you have visual data representation (e.g., graphs or tables), ensure they are properly labeled and referenced in the text. Each figure/table should have a clear legend or caption explaining what it represents. Comments: Addressed.

Reviewer #2: The manuscript is under journal scope but need to some modifications should be addressed as follows:

1. Provide a detailed justification for selecting these classifiers, including a comparison with alternative deep learning or machine learning models. Highlight their strengths and relevance to the problem being addressed. Comments: Addressed in Section 1.3

2. Describe the dataset in detail, including its size, source, and characteristics.

3. Include confusion matrices, ROC curves, precision-recall curves, or SHAP (SHapley Additive exPlanations) analysis to provide deeper insights into model performance. Comments: Addressed in Results and Findings Section 3.1.

4. Update the abstract to include key performance metrics (e.g., accuracy, F1-score, AUC-ROC). Comments: Addressed.

5. Discuss the robustness of the system against adversarial attacks and propose strategies to enhance security. Comments: Addressed in Discussion of Findings under the Section 3.1

6. Refine the language to ensure clarity and readability. Comments: Addressed.

---

## [Editor Report · Decision Letter 2]

Adaptive DDoS Detection Mode in Software-Defined SIP-VoIP Using Transfer Learning with Boosted Meta-Learner

PONE-D-25-00389R2

Dear Authors

We’re pleased to inform you that your manuscript has been judged scientifically suitable for publication and will be formally accepted for publication once it meets all outstanding technical requirements.

Kind regards,

Dr. Taimur Bakhshi

Academic Editor

PLOS ONE

Additional Editor Comments (optional):

Thank you for addressing pervious comments, I am pleased to process this further.

---

## [Editor Report · Acceptance letter]

PONE-D-25-00389R2

PLOS ONE

Dear Dr. Ukwandu,

I'm pleased to inform you that your manuscript has been deemed suitable for publication in PLOS ONE. Congratulations! Your manuscript is now being handed over to our production team.

Kind regards,

on behalf of

Dr. Taimur Bakhshi

Academic Editor

PLOS ONE